META-RESEARCH

# How parenthood contributes to gender gaps in academia

**Abstract** Being a parent has long been associated with gender disparities in academia. However, details of the mechanisms by which parenthood and gender influence academic career achievement and progression are not fully understood. Here, using data from a survey of 7,764 academics in North America and publication data from the Web of Science, we analyze gender differences in parenthood and academic achievements and explore the influence of work-family conflict and partner support on these gender gaps. Our results suggest that gender gaps in academic achievement are, in fact, "parenthood gender gaps." Specifically, we found significant gender gaps in most of the measures of academic achievement (both objective and subjective) in the parent group but not in the non-parent group. Mothers are more likely than fathers to experience higher levels of work-family conflict and to receive lower levels of partner support, contributing significantly to the gender gaps in academic achievement for the parent group. We also discuss possible interventions and actions for reducing gender gaps in academia.

**XIANG ZHENG, HAIMIAO YUAN AND CHAOQUN NI\***

**\*For correspondence:**
chaoqun.ni@wisc.edu

**Competing interest:** The authors declare that no competing interests exist.

## Introduction

Gender disparity is prominent in academia, notably in science and engineering (*Ceci et al., 2014*). In the United States in 2019, for example, only 16% of scientists and engineers were women (*National Center for Science and Engineering Statistics, 2021*). Women are also underrepresented across all professorship ranks (*Chesler et al., 2010*; *Mason et al., 2013*), especially among full professors – in 2018 only 32.5% of full professors in the United States were women (*Colby and Fowler, 2020*). Studies show that, compared with men, women academics produce fewer papers (*Larivière et al., 2013*; *Paul-Hus et al., 2015*), receive fewer citations (*Caplar et al., 2017*; *Maliniak et al., 2013*), and have narrower collaboration networks (*Ductor et al., 2021*). Women are also less likely to receive project funding (*Ley and Hamilton, 2008*; *Witteman et al., 2019*) or prestigious awards (*Lunnemann et al., 2019*; *Meho, 2021*). These gendered differences are usually correlated and mutually reinforcing, contributing to lower career satisfaction and higher attrition rates for women in academia (*Xu, 2008*). Understanding variables and mechanisms underlying the gendered disadvantage for women is critical for achieving gender equality in academia.

Parenting is considered highly related to gender disparities in academia (*Hunter and Leahey, 2010*; *Kelly and Grant, 2012*; *Morgan et al., 2021*; *Powell, 2021*). It contributes to the gender gap in many key research performance metrics, such as scientific productivity (*Mairesse et al., 2019*; *Morgan et al., 2021*), citations (*Lawson et al., 2021*), and academic collaboration (*Hunter and Leahey, 2010*). Evidence also shows that career satisfaction in academia is lower for mothers than fathers (*Beckett et al., 2015*), likely due to the reasons such as limited opportunities for tenure and promotion (*Finkel and Olswang, 1996*; *Mason et al., 2013*), lower salary (*Kelly and Grant, 2012*), and higher levels of work-family conflict (*Martins et al., 2002*). The prominent motherhood penalty could impel women to self-select away from academic careers and to leave such careers at higher rates than men (an effect known as the "leaky pipeline": *Anders, 2004*). Yet, our present knowledge about parenting and gender disparities in academia has been shaped primarily by studies that are university-wide (e.g., *Misra et al., 2012*), monodisciplinary (e.g., *Beckett et al., 2015*), and thus of limited sample scales. The size, type, and mechanism of the association between parenting and academic achievements need further examination on a larger scale.

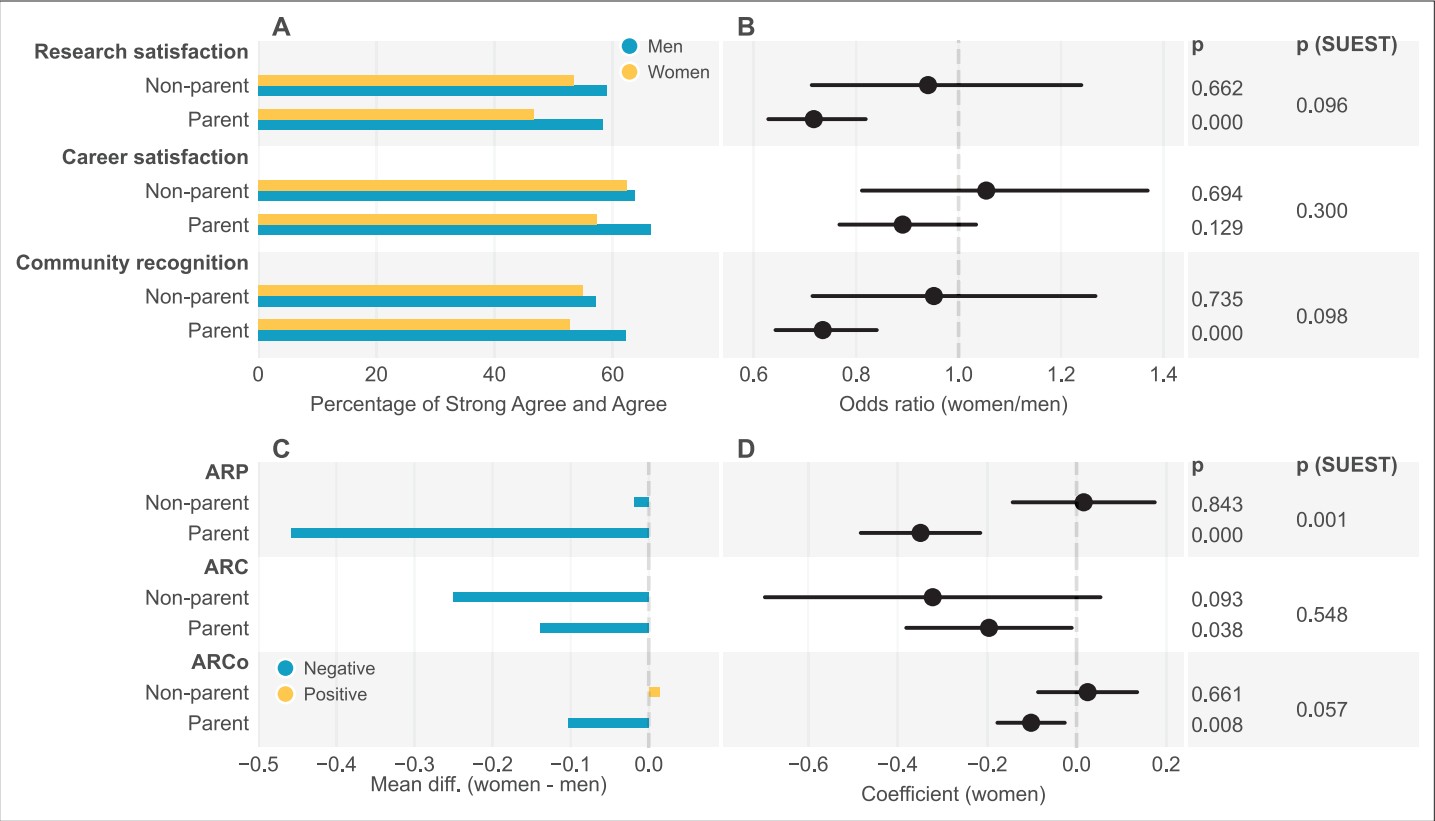

**Figure 1.** Subjective and objective career achievements by gender and parenthood status. (**A**). Percentage of satisfaction over research, career, and recognition by scholarly communities; (**B**). Women/men odds ratio for subjective career achievements; Control variables include discipline, career stage, partner job type, and race. Standard errors were clustered at the institution level. p (SUEST) values compare the odds ratio values between the parent and non-parent group; (**C**). Women-men difference in annual relative publication (ARP), average relative citation (ARC), and annual relative coauthor (ARCo). Positive values indicate female dominance and negative male dominance; (**D**). Coefficients for gender (women) based on linear regression analysis on ARP, ARC, and ARCo.

One potential reason parenting contributes to the gender disparity in academic careers is the work-family conflict. Workplace and family are both demanding in terms of time, energy, and loyalty, thus likely interfering with each other (*Fox et al., 2011*). Work-family conflict can be categorized into three types: (i) time-based conflict is the competition of time between different roles, e.g., mother vs. researcher; (ii) strain-based conflict happens when the strain in one role interferes with one's ability to perform in another role; (iii) behavior-based conflict happens when the behavior requirements as a role become incompatible with the behavioral expectation of another (*Greenhaus and Beutell, 1985*). Studies have indicated that women usually experience higher levels of work-family conflict than men due to parenting (*Martins et al., 2002*; *van Daalen et al., 2006*; *Wang et al., 2020*), and academia is no exception (*Fox et al., 2011*). Mothers who are academics spend more time on childcare responsibilities and household tasks (*Misra et al., 2012*)

and experience more research pressure and job stress than men (*van Daalen et al., 2006*). It is indicated that high work-family conflict can significantly affect women's career satisfaction, regardless of their age (*Martins et al., 2002*). However, few studies have focused on how varying levels of work-family conflict across parenthood status contribute to the gender gaps in academia regarding different forms of career achievements.

The role of family support should also be considered. Family has become "the newest battlefront in the struggle for gender equality" (*Hauser, 2012*), enduring and consequential for well-being across the life course (*Thomas et al., 2017*). Family support, especially partner support, represents a significant form of family-to-work enrichment and has positive effects (*Greenhaus and Powell, 2006*; *Kinnunen et al., 2006*). Partner support can be instrumental or emotional (*Adams and Golsch, 2021*; *Greenhaus and Powell, 2006*) and may decrease work turnover

**Table 1.** Child impact on career for parents.
Odds ratio (women/men) values were based on ordinal logistic regression. Standard errors were clustered at the institution level. Control variables include discipline, race, and type of partner job.

| | All | | Trainee | | Early career | | Middle career | | Late career | |
|---|---|---|---|---|---|---|---|---|---|---|
| | Women | Men | Women | Men | Women | Men | Women | Men | Women | Men |
| Negative (%) | 71.3 | 48.6 | 79.3 | 60.2 | 79.1 | 66.4 | 75.5 | 59.1 | 60.9 | 37.3 |
| Neutral (%) | 14.7 | 26.2 | 9.6 | 21.2 | 9.7 | 16.5 | 13.1 | 19.1 | 20 | 33 |
| Positive (%) | 14.1 | 25.2 | 11.1 | 18.6 | 11.2 | 17.1 | 11.4 | 21.9 | 19.2 | 29.7 |
| N | 3,105 | 2,530 | 208 | 118 | 618 | 333 | 1,172 | 745 | 1,107 | 1,334 |
| OR, 95% CI, p-value | 0.46 [0.41,0.51], *P*=0.000 | | 0.34 [0.20,0.61], *P*=0.000 | | 0.52 [0.39,0.70], *P*=0.000 | | 0.47 [0.40,0.56], *P*=0.000 | | 0.43 [0.37,0.51], *P*=0.000 | |

rate and boost women's perceptions of gains in productivity and career satisfaction (*Ferguson et al., 2016*; *Juraqulova et al., 2015*; *Watanabe and Falci, 2016*). Partner support can also relieve the tension inside the family and reduce the pressure from the family, mitigating work-family conflict and increasing well-being and career performance (*Dickson, 2020*; *Ferguson et al., 2016*; *Thorstad et al., 2006*; *Wang et al., 2020*). Although partner support has been found to help reduce work-family conflict – with the reduction for mothers being greater than that for fathers – the demand for partner support from mothers is often less satisfied (*Adams and Golsch, 2021*; *Dickson, 2020*). Despite previous discussions on the role of partner support in family-to-work enrichment, limited attention has been paid to the relationship between partner support and the gender disparity in academia.

Using data from the Web of Science and responses from 7,764 academics in the United States and Canada to a survey distributed in 2019 (see *Supplementary file 1*), this study examines the gender gaps in various measures of academic achievement (both objective and subjective), and how parenting-related work-family conflict and partner support mediate these gaps. Objective career achievements are observable, socially recognized indicators signaling one's human capital values (*Valcour and Ladge, 2008*). We use bibliometric indicators of scientific productivity, citation, and collaboration to measure objective career achievements (see Materials and materials). Subjective career achievements are one's subjective feelings of career attainments (*Ng et al., 2005*), including self-reported research satisfaction, career satisfaction, and the

perceived recognition by scholarly communities in this study.

Based on self-reported gender identification, the sample contains 4,425 (57.0%) women, 3,311 (42.7%) men, and 28 respondents (0.4%) who self-identified as non-binary. Due to the limited sample size for the non-binary group, our subsequent analyses only focus on women and men, which we admit is a limitation of our study. To better understand the role of parenting, we aggregated respondents based on their self-reported parenthood status: the parent group (n=5,670, 73.3%) and non-parent (n=1,534, 19.8%) group. Subsequent analyses refer to respondents in the parent group who self-identified as women and men as mothers and fathers, and those in the non-parent group as non-mothers and non-fathers. It should be noted that the two groups only include respondents who have ever married or cohabited (for two years or longer), given work-family conflict and partner support being variables of significant interest in this study.

## Results

### Parenthood and its overall compatibility with academic careers

Our results show significant gender differences in the parenthood status of academics. Across all disciplines, career stages, and races, a large majority (73.7%) of the respondents have at least one child. However, women (71.4%) are less likely than men (76.7%) to have children (OR = 0.82, 95% CI [0.73, 0.91], *P*=0.000; see Table S6 in *Supplementary file 2*) and are more likely to have fewer children (Tobit coefficient = −0.14, 95% CI

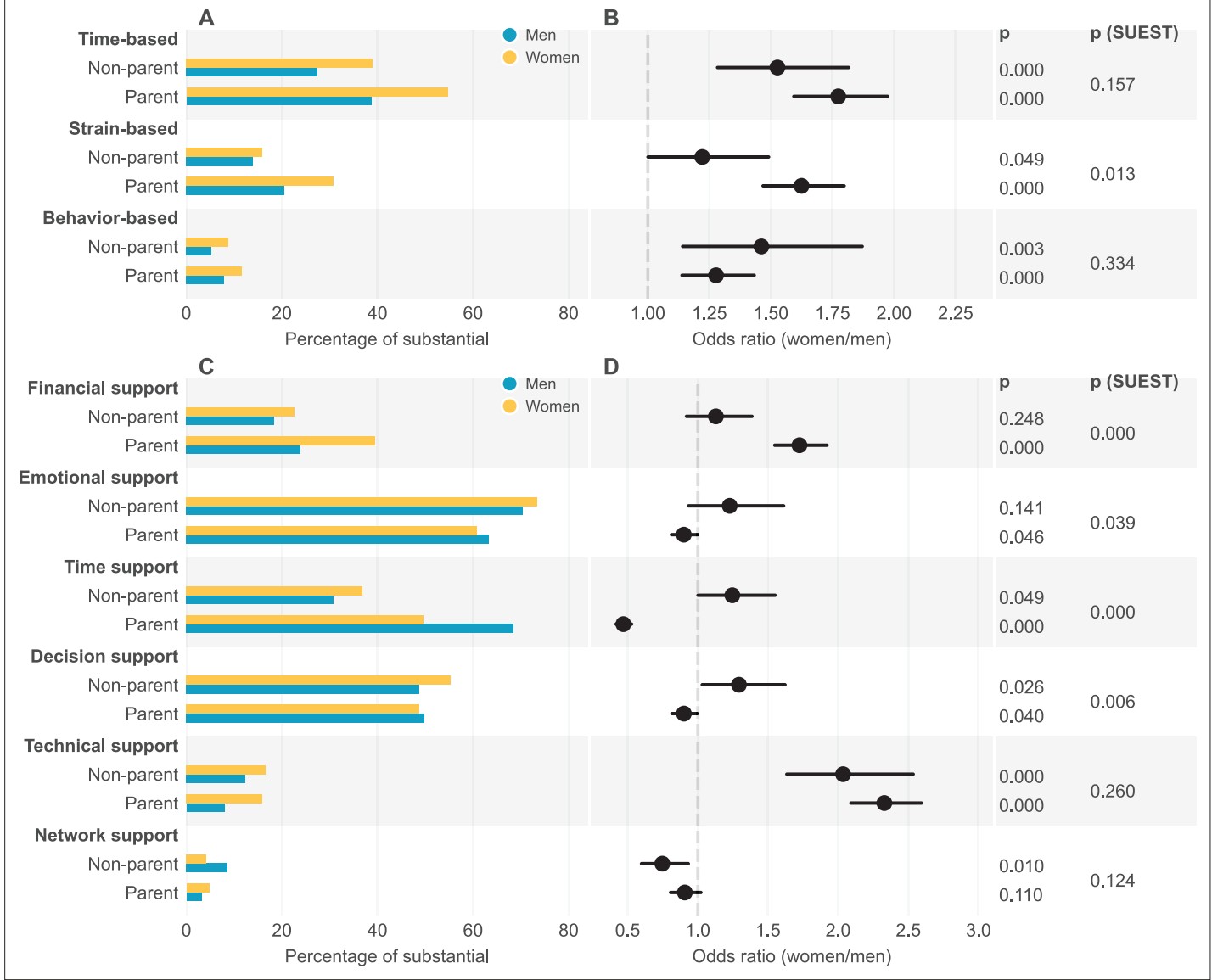

**Figure 2.** Forms of work-family conflict experienced, and partner support received by gender and parenthood status. (**A**) Percentage of women and men experiencing "substantial" conflict; (**B**) Odds ratio (women/men) for experiencing work-family conflict; (**C**) Percentage of women and men receiving "substantial" partner support; (**D**) Odds ratio (women/men) for receiving partner support. The (women/men) odds ratio values were based on logistic regression. Control variables include discipline, career stage, partner job type, and race. Standard errors were clustered at the institution level. SUEST was used to compare the odds ratio values between parent and non-parent group.

[-0.22,–0.07], *P*=0.000). This trend is different from the general population in the United States, where the percentage of adult women who have children is higher than that of adult men, and the average number of children mothers have is higher than that of fathers (***Monte, 2017***; ***National Center for Health Statistics, 2021***). Our results further show that the gender difference in parenthood status of academics is highly related to career considerations: For those with children, women are more likely than men to report that the number of children they have is related to

career considerations (OR = 2.34, 95% CI [2.08, 2.63], *P*=0.000). Among those without children, women (59.9%) are more likely than men (43.2%) to report that career considerations played a role in their parenthood status (OR = 2.10, 95% CI [1.69, 2.62], *P*=0.000; see Table S7 in ***Supplementary file 2***), while controlling for discipline, career stage, and race.

The gender difference in parenthood status in academia is likely due to the different perceptions of parenting compatibility with academic careers by women and men: Parenthood is perceived as

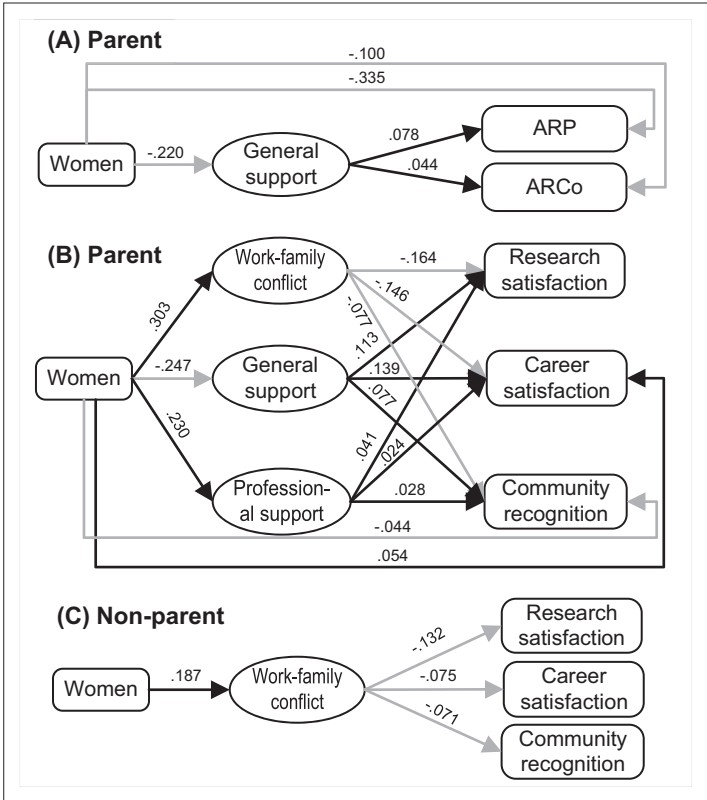

**Figure 3.** Mediation effect analysis models of partner support and work-family conflict between gender and subjective and objective career achievement measures. Only paths with statistically significant effects (p<0.05) are shown. Black and gray lines denote positive and negative mediating coefficients, respectively. (**A**) sets the objective career achievement as the outcomes and tests with the parent group (n=4,173). (**B**) sets the subjective career achievement measures as the outcomes and tests with the parent group (n=4,557). (**C**) sets the subjective career achievement measures as the outcomes and tests with the non-parent group (*n*=1,152).

less compatible with academic careers by women than by men (see *Table 1*). Women are more likely than men to report a negative impact on their career due to children (OR = 0.46, 95% CI [0.41,0.51], *P*=0.000). More specifically, 71.3% of mothers reported a negative child impact on their career, while only 48.6% of fathers indicated so. In contrast, 25.2% of fathers reported a positive impact on their career due to children, while only 14.1% of mothers indicated so. When aggregated by career stage (see Table S4 in *Supplementary file 1* for the categorization), women are more likely to report more negative child impact during each career stage: trainee (OR = 0.34, 95% CI [0.20,0.61], *P*=0.000), early career (OR = 0.52, 95% CI [0.39,0.70], *P*=0.000), middle career (OR = 0.47, 95% CI [0.40,0.56], *P*=0.000), and late career (OR = 0.43, 95% CI [0.37,0.51], *P*=0.000).

With limited generalizability due to insufficient sample size, we found that, of the 28 non-binary participants in our survey, 18 (64.3%) have been ever committed to a partnership. Seven of them

have children, among which six (85.7%) indicated that having children has a negative impact on their career development, and all seven (100%) respondents reported career considerations played a role in the number of children they have.

### Gender gaps in subjective career achievements

We operationalized the measurement of subject career achievements using respondents' self-reported satisfaction over research, career, and recognition by scholarly communities. Our results show that women academics, in general, are attached with lower levels of subjective career achievement than their male counterparts (see Table S8 in *Supplementary file 2*). Specifically, women are less likely than men to be satisfied with their research (OR = 0.76, 95% CI [0.68,0.85], *P*=0.000) and feel recognized by their scholarly communities (OR = 0.77, 95% CI [0.68,0.87], *P*=0.000).

Yet, the gendered differences vary by parenthood status. For the non-parent group,

women and men show no significant differences in research satisfaction (OR = 0.94, 95% CI [0.71,1.24], *P*=0.662), career satisfaction (OR = 1.05, 95% CI [0.81,1.37], *P*=0.694), or the recognition by scholarly communities (OR = 0.95, 95% CI [0.71,1.27], *P*=0.735). For the parent group, however, the gender gap is prominent: mothers are less likely than fathers to be satisfied with their research achievements (OR = 0.72, 95% CI [0.63, 0.82], *P*=0.000), and the recognition from their scholarly communities (OR = 0.73, 95% CI [0.64, 0.84], *P*=0.000). While research satisfaction is significantly different between mothers and fathers, our results show no significant gender difference in the career satisfaction of parents (OR = 0.89, 95% CI [0.77,1.03], *P*=0.129).

The seemingly unrelated estimation (SUEST) analysis further shows that there is no significant difference in the career satisfaction gender gaps between parents and non-parents (*P*=0.300). Furthermore, of those women who expressed satisfaction over their career achievements, 13.6% were not satisfied with their research achievements. The number is 8.9% for men (see Table S9 in *Supplementary file 2*). Research, teaching, and service being the three primary duties for many tenure-track faculty, it has been found that teaching and service are more heavily loaded on women than on men (*Bellas, 1999*; *Guarino and Borden, 2017*). If teaching and service are indeed loaded heavier on women, then mothers' lower satisfaction rates on research could be related to the extra amount of time and effort they put on these aspects, on top of mothers' already reduced working time due to parenting.

For the non-binary respondents, five (71.4%) out of seven parents are not satisfied with their research achievements, compared with four (36.4%) out of 11 non-parents. For career satisfaction, four (57.1%) out of seven parents expressed disagreement, compared with two (18.2%) out of 11 non-parents. For scholarly recognition, one (16.7%) out of six parents expressed disagreement, compared with two (18.2%) out of 11 non-parents. Yet, the results should be interpreted with limited generalizability due to the limited sample size.

## Gender gaps in objective career achievements

We operationalized the measurement of objective career achievements for academics using publication-based indicators, given the "currency" role that scientific publications play in the academic reward system (*Fogarty, 2009*).

The publication-based indicators used in this study include annual relative publication (ARP), average relative citation (ARC), and annual relative coauthor (ARCo). These indicators are used as measures for research productivity, citation, and the extent of collaboration, as normalized by discipline and time. The normalization was performed due to the varying scholarly practices across disciplines and the cumulative advantage in research achievements (such as citations) over time (see Materials and methods).

Our results further confirmed gender disparities in objective career achievements (See *Figure 1* and Table S10 in *Supplementary file 2*). Regardless of parenthood status, women are outperformed by men in all three measures of objective career achievement. Specifically, women's ARP is 0.28 (95% CI [–0.39,–0.17], *P*=0.000) units lower than men, women's ARC is 0.21 (95% CI [–0.38,–0.05], *P*=0.013) units lower than men, and women's ARCo is 0.08 (95% CI [–0.14,–0.01], *P*=0.023) unit lower than men. This echoes previous findings that women produce fewer publications (*Astegiano et al., 2019*; *van den Besselaar et al., 2017*), receive lower citations (*Larivière et al., 2013*; *Maliniak et al., 2013*), and have fewer coauthors (*Ductor et al., 2021*) than their male counterparts.

Yet, our results suggest parenthood may have played a significant role in the observed gender gaps in productivity, citation, and the extent of collaboration. In the parent group, the ARP, ARC and ARCo for mothers are 0.35 (95% CI [–0.48,–0.22], *P*=0.000), 0.20 (95% CI [–0.38,–0.01], *P*=0.038), and 0.10 units (95% CI [-0.18,–0.03], *P*=0.008) lower than those for fathers. However, in the non-parent group, women and men do not differ significantly in any of the three measures. Furthermore, the mean productivity difference between women and men is larger in the parent group than in the non-parent group: our SUEST analysis shows there is a significant difference between the gender coefficients (for ARP) of the parent and non-parent group regression analyses, indicating significant differences between productivity gender gaps between the parent and non-parent group (*P*=0.001). However, we do not observe such differences in citation and collaboration. These findings suggest that many previous findings on gender disparities in research productivity are likely due to the gender differences among parent academics, given that they are the majority of academics (73.7% of all academics in our study). Moreover, recognizing that scholarly practices may vary as scholars advance through their careers, we further

aggregated respondents by career stage (see Table S10 in *Supplementary file 2*). Results show that mothers produced fewer publications than fathers across early, middle, and late careers. In the meantime, women and men in the non-parent group do not vary significantly in productivity during any career stage.

Additionally, the mean values of ARP and ARC are generally higher for the parent group than for the non-parent group for both genders (see Table S10 in *Supplementary file 2*). Recognizing that the direct comparison of means ignores the potential skewness in data distribution and the effect of control variables, we further performed regression analysis to compare parents with non-parents (see Table S11 in *Supplementary file 2*). We found no significant difference between mothers and non-mothers in ARP, ARC, or ARCo. However, fathers have higher ARP and ARCo values than non-fathers, but not ARC. The finding of higher ARP for fathers than non-fathers is consistent with *Morgan et al., 2021*. Yet, we did not find such a pattern between mothers and non-mothers.

Overall, our results reveal the potential existence of both "motherhood penalty" and "fatherhood premium" (*Kelly and Grant, 2012*), and their relationships with gender gaps in objective career achievements. On the one hand, higher values of ARP and ARCo for fathers compared to non-fathers suggest the potential existence of the "fatherhood premium" for men. While reasons for such fatherhood premium require further investigation, we consider the relationship between the ARP values of fathers and non-fathers being related to their collaboration pattern, given the potential positive relationship between collaboration and productivity (*Larivière et al., 2015*). On the other hand, we found mothers and non-mothers show do not differ significantly in ARP. *Morgan et al., 2021* showed that annual productivity of academics tends to grow over the years in computer science, business, and history, while the annual productivity of women grows slower or even stagnates after the first birth of their first child. Our findings suggest that such motherhood penalties may offset the productivity growth in the long run, resulting in the observed gender gaps in productivity. Therefore, we consider that fatherhood premium and motherhood penalty co-contribute to the gender gaps in the objective career achievements of parent academics, along with many other factors not discussed in this study.

For non-binary respondents, the ARP is 1.34 for parents and 1.36 for non-parents, the ARC is 1.58 for parents and 1.79 for non-parents, and the ARCo is 0.92 for the parents and 0.98 for the non-parents. The limited descriptive analysis shows that parenthood may also be related to the career achievements of the non-binary respondents. However, given the small sample size, the above comparisons may lack statistical power and therefore be interpreted with limited generalizability.

### Gender gaps in work-family conflict and partner support

When asked about factors that impeded their career, both genders agreed that work-family conflict is the major obstacle: 71.8% of women and 68.0% of men indicated they experienced "a little" to "substantial" levels of work-family conflict that impeded their career development, regardless of their parenthood status. However, the gender gap in work-family conflict is more significant in the parent group than in the non-parent group: Our results show that mothers are more likely than fathers to experience higher levels of work-family conflict (OR = 1.31, 95% CI [1.19,1.45], $P$=0.000), but this does not hold for non-parent academics (OR = 1.05, 95%CI=[0.85,1.28], $P$=0.662) (see Table S12 in *Supplementary file 2*). Additionally, work-family conflict may occur in various forms related to childbearing and childrearing. When work-family conflict is categorized into the three types of conflicts, our results show that mothers are more likely than fathers to experience higher levels of time-based (OR = 1.77, 95% CI [1.59,1.97], $P$=0.000), strain-based (OR = 1.62, 95% CI [1.47,1.80], $P$=0.000), and behavior-based (OR = 1.28, 95% CI [1.14,1.43], $P$=0.000) work-family conflict. Meanwhile, the levels of conflict in all three forms are higher for parents than for non-parents, as expected. We also observed significant gender gaps in the non-parent group regarding time-based (OR = 1.53, 95% CI [1.28,1.82], $P$=0.000), strained-based (OR = 1.22, 95% CI [1.00,1.49], $P$=0.049), and behavior-based (OR = 1.46, 95% CI [1.14,1.87], $P$=0.003) conflict, indicating that women are more likely to experience these conflicts than men even without children.

On the other hand, we should not overlook the role of partner support, which is beneficial for career success in many areas. We asked about the levels of partner support for careers received by academics, including financial, emotional, time, decision, technical (support for one's productive activities with personal skills, techniques,

and expertise), and network support. Our results show that, women academics are overall more likely than men to receive higher levels of financial (OR = 1.55, 95% CI [1.41, 1.71], P=0.000) and technical support (OR = 2.28, 95% CI [2.06, 2.53], P=0.000), but less likely to receive higher levels of time (OR = 0.58, 95% CI [0.52, 0.64], P=0.000) and network support (OR = 0.87, 95% CI [0.78, 0.97], P=0.012) from their partners. Yet, the gender difference in partner support, similar to that in work-family conflict, also varies by parenthood status. Mothers are less likely than fathers to receive higher levels of time support (OR = 0.47, 95% CI [0.42, 0.53], P=0.000), a pattern that does not hold in the non-parent group. Additionally, mothers are more likely than fathers to receive more financial support (OR = 1.72, 95% CI [1.55, 1.92], P=0.000), while non-mothers are not (OR = 1.13, 95% CI [0.92,1.39], P=0.248).

Mothers are also less likely than fathers to receive higher levels of decision support (OR = 0.90, 95% CI [0.82, 1.00], P=0.040) and less likely to receive higher levels of emotional support (OR = 0.90, 95% CI [0.81,1.00], P=0.046). Women are more likely than men to receive higher levels of technical support in both the parent (OR = 2.33, 95% CI [2.09,2.59], P=0.000) and non-parent (OR = 2.03, 95% CI [1.63,2.53], P=0.000) group (see *Figure 2* and Table S13 in *Supplementary file 2*). We further found that the gender difference in received support varies significantly by respondents' parenthood status: The women to men odds ratios for the parent group vary significantly from that for the non-parent group regarding financial (P=0.000), emotional (P=0.039), time (P=0.000), and decision support (P=0.006).

### The mediation effect of work-family conflict and partner support

While the above findings suggest the gender gaps in academics' objective and subjective career achievements, it is unclear what roles work-family conflict and partner support play in forming these gaps. We then used mediation effect analysis to unveil the possible mediating role of work-family conflict and partner support. To increase the interpretability of variables, we first extracted four factors from the three types of work-family conflict and six forms of partner support (nine variables in total) based on principal component analysis and varimax rotation, with each attached with at least one variable contributing significantly (factor loading ≥0.5) to the factor. The four factors explained 66.69% of the total variance. The labels for the four factors

are work-family conflict (including time-, strain- and behavior-based conflict), financial support, professional support (including technical and network support), and general support (including emotional, decision, and time support, all of which do not require special skills or economic power) (see Table S6 in *Supplementary file 2*).

The mediation effect analysis based on the extracted factors shows that work-family conflict and partner support are significant mediator variables contributing to the association between gender and objective and subjective career achievement measures for parents, albeit with different underlying mechanisms (see *Figure 3* and Table S15 in *Supplementary file 2*). General support is a mediating variable for the association between gender and parents' objective career achievement. Mothers received less general support than fathers, which contributes significantly to mothers' lower levels of ARP (Path effect [PE]=−0.017, 95% percentile CI [-0.031,−0.004]) and fewer ARCo (PE = −0.010, 95% percentile CI [-0.018,−0.002]). For the association between gender and subjective career achievement measures, mothers are subject to higher levels of work-family conflict than fathers, which leads to lower levels of research satisfaction (PE = −0.050, 95% percentile CI [-0.062,−0.038]) and scholarly community recognition (PE = −0.023, 95% percentile CI [-0.031,−0.016]).

The significantly less general support received by mothers are also shown to disadvantage mothers in research satisfaction, career satisfaction, and scholarly community recognition (PE = −0.028, 95% percentile CI [-0.037,−0.019]; PE = −0.034, 95% percentile CI [-0.045,−0.025]; PE = −0.019, 95% percentile CI [-0.026,−0.013]). On the contrary, mothers receive more professional support, which mitigates the negative association between gender and the three measures, albeit with comparatively small indirect effects (PE = 0.009, 95% percentile CI [0.003, 0.016]; PE = 0.006, 95% percentile CI [0.000, 0.011]; PE = 0.006, 95% percentile CI [0.001, 0.012]).

For non-parent academics, the mediating effect of work-family conflict is also observed between the association of gender and subjective career success: women are subject to higher levels of work-family conflict than men, inhibiting women's satisfaction over research satisfaction (PE = −0.025, 95% percentile CI [-0.045,−0.008]), career satisfaction (PE = −0.014, 95% percentile CI [-0.028,−0.004]), and scholarly community recognition (PE = −0.013, 95% percentile CI [-0.026,−0.003]). Work-family conflict and partner support show no significant mediation effect

on the associations between gender and the objective career achievement for the non-parent group.

## Discussion

Our analysis of the parenthood status of academics provides new insights into the dilemma many women academics face: motherhood or academic career. On the one hand, mothers who are academics are significantly more likely than fathers to report higher levels of negative impact on careers due to children, reflecting the motherhood penalty on the academic career (*Bonache et al., 2022*; *Misra et al., 2012*). The perceived negative impact of parenting could intensify gender inequalities, as previous research found it was one of the major systemic barriers pushing women to self-select away from academia, contributing to the academic "leaky pipeline" (*Anders, 2004*). On the other hand, we found that women are less likely than men to have children and have fewer children.

The insignificant differences between non-fathers and non-mothers in their productivity, citation, and collaboration imply that parenthood contributes to the gendered differences in these three aspects for parents. Given the observed motherhood penalty, it is unsurprising that academic women have fewer children and are less likely to have children than men, which our results suggested as being related to career considerations: Our results show non-mothers are more likely than non-fathers to report their decision to be childfree was related to career considerations, and that mothers are more likely than fathers to report that the lower number of children they have was also related to career considerations.

Our study also unveils gender gaps in both objective and subjective career achievements of academics and suggests much of the gender gap only exists in the parent group. Specifically, we find gender differences in all measures of subjective career achievement and objective career achievement for the parent group. For the non-parent group, however, no significant gender difference was found in any career achievement measures. We thus argue that given many previous studies (*Bendels et al., 2018*; *Caplar et al., 2017*; *Holliday et al., 2014*; *Larivière et al., 2013*) reported overall gender disparities in academia without differentiating between parents and non-parents, these disparities may primarily derive from the differences between academic mothers and fathers. Follow-up studies

on gender disparities could further examine this issue by comparing parents and non-parents.

Given the role of productivity in the academic reward system, current policies and regulations that aim to provide short-term support for child-caring and recovery right after birth or adoption are insufficient to bridge the gender gap in academia. Furthermore, the impact of parenting on the research output of mothers may last longer than asserted by previous studies (*Morgan et al., 2021*): We find significant gender differences in productivity across all career stages among the parents, but no such differences have been observed among the non-parents. This suggests that policies and interventions addressing gender inequalities in the scientific workforce should go beyond short-term assistance such as parental leaves and extend long-term support for mothers. Other forms of sustainable family-friendly support, such as subsidized childcare, onsite childcare, flexible working schedules, and supportive working environments, are available options (*Feeney and Stritch, 2019*). Additionally, given the observed gender gaps in research productivity, the current metric-based evaluation, especially those based on productivity, will undoubtedly hurt mothers more. Therefore, additional metrics and qualitative evaluations considering parenthood and individual conditions would be critical to reducing gender inequalities in academic settings.

Our findings on mothers' higher chances of encountering more elevated levels of work-family conflict in all the forms of time-, strain-, and behavior-based conflict confirm that mothers indeed suffer more from multiple social roles: mother and researcher. Previous findings suggested that mothers are children's primary caregivers, requiring extra time and engagement in parenting activities (*Dickson, 2020*). We also found that mothers are significantly more likely than fathers to experience time-based conflict and less likely to receive the time, emotional, and decision support from partners, a dangerous signal for the family to sustain marital satisfaction and gender equality (*Mickelson et al., 2006*; *Thorstad et al., 2006*). On the contrary, mothers are more likely to receive financial and technical support from their partners. Mothers' higher likelihood of receiving financial support might be related to the entrenched gender division of household labor, where fathers are more likely to be the breadwinners and bear financial responsibilities (*Hauser, 2012*).

Our mediating effect analysis results provide new evidence for the claim that family

is "the newest battlefront in the struggle for gender equality" (*Hauser, 2012*) for academia, confirming that family-related reasons are significantly associated with gender gaps in career achievements of parents. Specifically, mother who are academics receive less general support (including time, emotional and decisional support) from their partners than fathers, contributing to the gender gap in productivity and collaboration. It corroborates the finding that shortened working time and limited decision support for more work engagement restrict mothers' opportunities for networking and academic collaboration (*Finkel and Olswang, 1996*; *Mason et al., 2013*). The limited general support from partners thus may hurt the research productivity of mothers directly or indirectly by constraining research collaborations, which usually leads to higher productivity (*Abbasi et al., 2011*; *Sonnenwald, 2007*). Moreover, the higher levels of work-family conflict experienced by mothers also worsen the gender gap in satisfaction over research and perceived recognition by parents, which are highly related to women's turnover intention (*Watanabe and Falci, 2016*). Reducing mothers' work-family conflict and increasing their general support could thus reduce the gender gap in academics' perceived career achievement. These mediating variables indicate the directions for mitigating the negative impacts of motherhood: Partner support in the forms of time, emotional, and decision support, and other collective efforts to reduce the time-based, strain-based, and behavior-based conflict for mothers.

By combining large-scale survey and bibliometric data, we reveal the decisive role of parenthood in the current gender gap in academia and the underlying mechanisms of how parenting-related conflicts and support mediate the gap. Previous studies asserted the importance of various forms of support from governments and institutions (*Morgan et al., 2021*). Yet, our results show that family is another new battleground: assistance and support provided by partners in time, emotional and decisional support are also vital. Addressing gender inequality in academia is a task requiring collective intelligence. In addition to parenting-related support by governments and institutions, which could be sometime inadequate due to restrictions on resources and regulations in the U.S., the support from partners and families will also help narrow the gender gap in academia.

## Limitations

Like many other survey-based studies, our research is not immune to potential bias caused by the self-selection of respondents. Given the topic of our survey, it is not surprising that a higher proportion of women than men in the population may have responded (see Table S6 in *Supplementary file 2*). Future efforts on this topic should consider strategies to encourage more responses from men academics to ensure the representativeness of both genders. We also restricted our survey to be sent to individuals associated with institutions in the U.S. or Canada only, considering the similarity between the "laddered" academic systems between the two countries and their distinctions from other countries and regions. But this omitted the rising research power in other countries and areas and restricted the generalizability of our findings to the scientific workforce in North America. Future research may examine this issue in other countries and regions of the world to understand the gender gap in the global scientific workforce.

The current study is also limited by the fact that the survey did not collect data to distinguish same-sex and opposite-sex partners, where gender roles and expectations may differ. Additionally, the current study lacks data on the types of childcare responsibilities by parents, such as whether they are custodial parents or non-custodial parents and whether children live with them. To further understand how childcare responsibilities contribute to gender gaps in academia, data and analyses based on the aforementioned factors will be critical for understanding the true mechanisms and possible policy implications for addressing gender inequalities in the scientific workforce.

Furthermore, we operationalized objective career achievement in productivity, citation, and the extent of collaboration as metrics of publication, citation, and coauthor. Yet, publication practices of academics only account for a part of their career achievements, and other measures such as grant funding and prestigious award have been overlooked. Moreover, it is known that disciplines relying on venues other than journals are underrepresented in the Web of Science (which largely indexes journal publications). Therefore, we may underestimate respondents' productivity, citation, and collaboration within those disciplines. The strategy we used to normalize these measures by field helps mitigate this issue, given we calculated respondents' measures of productivity, citation, and collaboration proportional to

fellow researchers in the same disciplines. Finally, the cross-sectional questionnaire data from our study restricted the potential to explore time-series change. To analyze the causal relationship between parenting and academics' career achievements, we plan to track the respondents and organize similar surveys in the future, comparing the parents as the treatment group with non-parents as the control group within gender.

## Materials and methods

### Data collection

This study relies on two data sources: a large-scale survey distributed to 99,168 researchers and their publication profiles from the WoS database by Clarivate Analytics. For the survey, we extracted from WoS 396,674 researchers who published at least one paper from 2000–2019,, were affiliated with an institution located in the United States or Canada and had a valid email address associated with them. We then randomly sampled 99,168 researchers (25%) from the population and sent a survey with 53 questions about parenting and career development in 2019 through Qualtrics (see *Supplementary file 3*). A total of 10,333 respondents initiated the survey, of which 9,105 finished. An analysis of the attrition failed to identify a common point of departure, suggesting individual variability in dropout rather than failed survey construction. This study's final number of respondents is **7,764** after removing respondents lacking information for critical variables of interest (see Table S1 in *Supplementary file 1*). We also collected data for non-binary academics, who were excluded from our analysis due to an insufficient sample (n=28). The University of Iowa's Institutional Review Board approved the study (IRB No. 201901776). All respondents gave informed consent before participating in the survey.

To assess the objective career achievement of respondents, we extracted the bibliographic records for individuals, including productivity (for which our proxy was papers), citation (for which our proxy was citation), and extent of collaboration (for which our proxy was unique coauthors). Recognizing the difference in the publication practices across disciplines and the cumulative nature of these measures, we normalized the three indicators of objective career achievement by discipline and time (see *Supplementary file 1*). We then use the annual relative publication (ARP), average relative citation (ARC), and annual

relative coauthor (ARCo) as indicators for productivity, citation, and extent of collaboration.

### Statistical analysis

This study used binary logistic regression, ordinal logistic regression, linear regression, and Tobit regression model to explore the gender difference in academic careers, depending on the measurement scale of outcome variables. In logistic regression analysis, the odds ratio (OR) of gender (women over men) is computed to explore the role of gender in outcome variables. An OR value lower than 1 indicates women are less likely to produce the outcome than men. Linear regression analysis yields a coefficient for gender (women = 1 and men = 0) rather than odds ratio, where a value below (above) 0 indicates that being a woman has a negative (positive) impact on the outcome measure. We used the Tobit regression model to estimate the relationship between parents' gender and child number, which is censored as the survey took six as a threshold for child number.

A consistent list of variables was used as control variables across the study, including disciplinary area, career stage, partner job type, and race. We did not include the institution as one control variable because the respondents' current affiliated institution may also be the outcome of their previous publications and career satisfaction in other workplaces. Instead, we clustered the standard errors in regressions at the institution level, considering an unobserved part of the residual may be correlated within an institution (e.g., an institution created a high-intensity work culture that influenced its employees' publishing behaviors satisfaction). We also used seemingly unrelated estimation (SUEST) to compare if gender differences vary significantly across parents and non-parents (see *Supplementary file 1*). Observations with missing values were removed without imputation.

This study analyzed the mediating effect of work-family conflict and partner support in the association between gender and career achievements of academics using mediation effect analysis. Before performing the analysis, we first extracted four factors from partner support and work-family conflict variables using principal component analysis and varimax rotation. The Cronbach's alpha of each factor's principal variables is >0.6, showing their acceptable internal consistency within a factor (see Table S14 in *Supplementary file 2*). We constructed a mediation effect analysis model for all subjective and

objective career achievement measures to test the indirect effects mediated by the four factors from gender to the outcome measures for the parent group (see Table S15 in *Supplementary file 2*). A bootstrap sampling procedure was used with 5,000 iterations to compute 95% confidence intervals and statistical significances for all indirect effect paths—the estimation controlled for career stage, disciplinary area, race, and partner job type. Standard errors are clustered by academics' affiliations to account for the non-independence of observations in the same affiliation.

### Acknowledgements
We thank Observatoire des sciences et des technologies at the University of Quebec in Montreal for access to the Web of Science data. We also thank Dr. Erjia Yan (Drexel University) for insights and comments on earlier drafts of the manuscript. This paper is available open access thanks to the University of Wisconsin Information School's Sarah M Pritchard Faculty Support Fund.

**Xiang Zheng** is in the Information School, University of Wisconsin-Madison, Madison, United States
ⓘ http://orcid.org/0000-0002-6619-5504
**Haimiao Yuan** is in the College of Education, The University of Iowa, Iowa City, United States
**Chaoqun Ni** is in the Information School, University of Wisconsin-Madison, Madison, United States
chaoqun.ni@wisc.edu
ⓘ http://orcid.org/0000-0002-4130-7602

*Author contributions:* Xiang Zheng, Conceptualization, Data curation, Software, Formal analysis, Visualization, Methodology, Writing – original draft, Writing – review and editing; Haimiao Yuan, Conceptualization, Data curation, Software, Formal analysis, Investigation, Visualization, Methodology, Writing – original draft, Writing – review and editing; Chaoqun Ni, Conceptualization, Resources, Data curation, Software, Formal analysis, Supervision, Funding acquisition, Validation, Investigation, Visualization, Methodology, Writing – original draft, Project administration, Writing – review and editing

*Competing interests:* The authors declare that no competing interests exist.

*Ethics:* Human subjects: The survey of this study was approved by the IRB board at the University of Iowa (IRB ID#201901776) IRB-02 DHHS Registration # IRB00000100, Univ of Iowa, DHHS Federalwide Assurance # FWA00003007. Below is the consent information from the approved IRB: You are invited to participate in a research project being conducted at the University of Iowa regarding the career development of researchers. The primary purpose of this study is to investigate the relationship between marriage, parenthood, gender, and the career trajectories of researchers. We aim to understand whether, and to what degree, these factors are related to the professional development of researchers. This project will provide implications for future scientists about their work-life management and career development, as well as related stakeholders, for the purpose of creating a better environment that will facilitate the development of researchers' careers. If you agree to participate, we would like you to complete an online survey (found below). You are free to stop taking this survey if you prefer not to answer any question. It will take approximately 15 to 20 minutes. Confidentiality research data will be kept anonymous and secure (encrypted and stored in a locked file) for up to 10 years and will then be deleted. Taking part in this research study is entirely voluntary. If you do not wish to participate in this study, you are free to decline. You may also withdraw from this project at any time, without consequences or recrimination. You will NOT be asked for an explanation for your withdrawal. Should you choose to withdraw after finishing the survey, please advise the project manager or any member of the research team. In the case of early withdrawal from the study, data will be destroyed immediately. If you have any questions about this project, please contact Haimiao Yuan (haimiao-yuan@uiowa.edu) at the University of Iowa. If you have questions about the rights of research subjects, please contact the Human Subjects Office, 105 Hardin Library for the Health Sciences, 600 Newton Rd, The University of Iowa, Iowa City, IA 52242-1098, (319) 335-6564, or e-mail irb@uiowa.edu. Thank you very much for your consideration.

### Funding

| Funder | Grant reference number | Author |
| --- | --- | --- |
| Wisconsin Alumni Research Foundation | 135-AAI3865 | Chaoqun Ni |
| University of Wisconsin-Madison | Sarah M. Pritchard Faculty Support Fund | Chaoqun Ni |

The funders had no role in study design, data collection and interpretation, or the decision to submit the work for publication.

**Decision letter and Author response**
Decision letter https://doi.org/10.7554/eLife.78909.sa1
Author response https://doi.org/10.7554/eLife.78909.sa2

# Additional files

## Supplementary files

• MDAR checklist

• Supplementary file 1. Tables S1-S5; Survey procedures; Operationalization of key variables; Objective career achievement measures; Statistical analysis.

• Supplementary file 2. Tables S6-S15.

• Supplementary file 3. Survey questions used by the study.

## Data availability

All data needed to evaluate the conclusions in the paper are present here and in the supplementary material. Aggregated or de-identified data on variables used in this study is available on GitHub (https://github.com/UWMadisonMetaScience/parenting, copy archived at swh:1:rev:46416c03c83eb28b77f6f17650537b2b413da663).

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
