## [Decision Letter]

**Decision letter after peer review:**

Thank you for submitting your article "How parenthood contributes to gender gaps in academia" to *eLife* for consideration as a Feature Article. Your article has been reviewed by two peer reviewers, and the evaluation has been overseen by two members of the *eLife* Features Team (Helga Groll/Peter Rodgers). The following individual involved in review of your submission has agreed to reveal their identity: Diego Kozlowski.

The reviewers and editors have discussed the reviews and we have drafted this decision letter to help you prepare a revised submission.

Summary:

Zheng and colleagues present an interesting study based on analyses of a large-scale survey examining the relationship between parenthood and gender gaps in academia. This subject is of considerable relevance to the academic community and would be an appropriate thematic fit for *eLife*. However, the article needs significant improvement in clarity of language and the authors engage in quite a bit of conjecture that is not supported by their primary findings. Discussions of gender, parenting and career achievement all involve sensitive topics. Precision of language and staying within bounds of what is supported by the data are critical to clearly communicate the author's findings and avoid offense. This is a theme that will come up in many of the specific comments below.

The article overall shows a very thorough analysis, it is very well written, and has a careful design. It is an important contribution to the field, and clarifies very specific mechanisms of gender inequality. Below there are a number of recommendations and concerns. We think that clarifying those would help them to make the article even stronger.

Essential revisions:

1. Why do "All" parent categories have higher ARPs and ARCs (for both men and women) compared to nonparents (Table S10)? Do female parents produce more publications and receive more citations than female non-parents? Do male parents produce more publications and receive more citations than male non-parents? This seems like an important point to discuss and one which is at odds with the author's overall thesis. Also, as mentioned below, Morgan et al. 2021 report that faculty of both genders who have children are slightly more productive than those who do not. This point should be discussed and put into context with the author's data/conclusions.

2. I did not extensively math check, but in areas where I did, the numbers were confusing. Unless I am misunderstanding where the numbers in the text are coming from.

For example, on page 10,

– "women's ARP is 0.28 units lower than men": On the graph the numbers are 2.17-1.79 = 0.38?

– "women's ARC is 0.21 units lower than men": On the graph the numbers are 2.27-2.08 = 0.19?

Or on page 11,

– "ARP, ARC and ARCo are 0.35, 0.2 and 0.10 units…" On the graph the numbers are 2.27-1.82 = 0.45, 2.28-2.13 = 0.15 and 1.09-0.98 = 0.11?

3. The author's discuss the "motherhood penalty". There is also a well-documented "fatherhood premium." How does this fit in with the author's results? Do the author's see evidence of a "fatherhood premium" in their data?

4. Morgan et al. (https://www.science.org/doi/10.1126/sciadv.abd1996) surveyed 3064 tenure-track faculty in 450 departments in the US and Canada. What advances are made in the current datasets/analyses? How do the results compare? In addition to the point mentioned above about faculty of both genders who have children being reported as more productive than those without, Morgan et al. also reports that results are inconclusive as to whether long-term publication rates are affected by parenthood. Can the authors place their study and results into context here?

5. Were there any differences for men/women parents/non-parents observed between different fields within academia? Are the conclusions the same regardless of discipline? Are the disparities the same regardless of discipline?

6. The authors do not distinguish between respondents who currently have childcare responsibilities (i.e., children < 18 currently residing with them) versus those who have ever had children (i.e., children > 18, non-custodial parents, etc.). The authors also do not distinguish between same-sex and opposite sex parents, where gender roles and expectations may differ. These should be raised in the Limitations section of the manuscript, as the current analyses focus on a model of a traditional, opposite-sex, nuclear family.

7. Data needs to be fully referenced within the text and shown if referenced. For example, on page 7 of the Results section, the authors refer to women in academia being more likely to have fewer children (presumably as compared with men?). There is not a reference for a data table here and the data does not appear to be in the supplemental tables? The authors then state "This trend is different from the US general population, where more adult women have children and have more children than men." (REF). Do the authors mean primary caretaking responsibilities of children or children that reside with them?

8. On page 11, last paragraph, the authors discuss results around career stage. This needs a reference to the data table.

9. On page 7, the authors state: "Among those without children, women (59.9%) are more likely than men (43.2%) to report that they chose to be childfree due to career considerations…". If I understand correctly, the survey question was: "Is the number of children (include 0) you currently have related to your career considerations (more or less)?" The interpretation is cause and effect (women chose to be childfree due to career) whereas the survey question asks if these were related considerations (which could include additional factors?).

10. In the Introduction, the authors use the term "non-binaries" to refer to individuals whose self-reported gender identity on the survey was "Other". Please consult the APA guidelines for bias free language: https://apastyle.apa.org/style-grammar-guidelines/bias-free-language/gender

11. In the Introduction, the authors state, "The multigenerational gender role beliefs, which characterize men and women with a simplistic set of features, may direct women academics toward women's normative roles in the family (REFs). Some women academics may become 'maternal gatekeepers" who are reluctant to relinquish family responsibility" (REFs). This seems to stray from the facts and into speculative territory not supported by the author's data.

12. In the Introduction, the authors state, "Although partner support is believed to be more critical to women's career success than to men, especially for those in parenthood, women's demand for partner support is less satisfied" (REFs). What does this mean? Women are more dependent on partner support for career success regardless of parenthood? Is this supported by data? Men need less partner support even in parenthood? Doesn't the author's data show male academic parents receive more time support, etc.?

13. This sentence between page 8-9 is confusing: "While the satisfaction over research is significantly different between mothers and fathers, our results show no significant gender differences in the career satisfaction of parents…therefore, it is likely that women consider more on other gains (e.g., emotional gains) from teaching and service while assessing their overall career satisfaction, while men emphasize research." What do the authors mean by "women consider more on other gains (e.g., emotional gains)" and is this data-driven or speculation? And what is meant by "men emphasize research"? It seems a stretch to attribute motivation / feelings to the author's data.

14. What does this sentence on page 13 mean, "women are usually disproportionately associated with technical work in research" (REF)?

15. In the Discussion section, the authors state, "the perceived negative impact of parenting is one of the major systematic barriers for women's self-selecting attrition from academia…" Can the authors support the claims of self-attrition and self-attrition due to a perceived negative impact of parenting? This seems to imply that women are deciding it's too hard to have children and an academic career, and then deciding to leave academia. Is there data to support this? This also entirely ignores bias/discrimination against women and mothers, which is well documented in academia.

16. In the Discussion section, while I don't believe the authors intended this statement to be offensive, some portions may come across that way: "Women in academia…tend to slough off the burden of children… Given the motherhood penalty, it is unsurprising that many women stay in academia by paying the price of being childfree…" In particular, this places value judgements around having children or not having children. It also seems unsupported by the data.

17. In the Discussion section, the authors discuss the finding that "mothers are more likely to receive financial and technical support from their partners." By stating "It reflects the entrenched gender division of household labor that expands with the new childcare task: Mothers are recognized, and even morally self-recognized, as the babysitters and fathers are the breadwinners" (REF). "As a social norm…fathers bear the financial responsibility. In contrast, fathers compensate mothers' loss due to extra childcare responsibility by offering financial and technical support." Again, can the authors draw this conclusion from their data?

18. In the Discussion section, the authors again make reference to "high attrition rates of mothers in the scientific workforce due to lower satisfaction rates." Can the authors backup the idea of higher attrition rates of mothers compared with non-mothers in the scientific workforce, and also that the reasons for attrition are due to lower satisfaction rates (i.e., self-selection)?

19. P.6. I am happy to see that the authors included non-binary people in their survey. I also understand that the small sample makes it impossible to consider NB into the regression models. Nevertheless, it would be very important to include their responses somehow. A qualitative analysis of the 28 responses from NB people after the quantitative analysis of men and women, to see where they locate within the more general results would be a very interesting complementary analysis.

20. It is not clear to me why the authors decided to send the survey to 25% of the population. If there was a technical limitation, it would be good to state it. Besides this, there is a 9% response rate. It would be helpful to see some robustness checks o whether that 9% is representative of the population or not. Some comparative statistics between respondents and non-respondents would help to disclose some potential self-selection biases.

21. The sample size of parents and non-parents is different, with a larger number of parents to non-parents (roughly 80% to 20%). Statistical measures are usually dependent on sample size, with larger sample sizes having more rejection power. Is it possible that the differences seen between groups, no significant bias on non-parents and significant bias on parents is just and statistical artifact of the sample sizes? The authors could try to check if this is the case using bootstrapped samples of equal size, or also consider directly the absolute values of the metrics to see if there is an important difference between the two groups -besides the statistical difference-.

Further on the previous point. On Figure 1A the authors show the different subjective satisfaction measures by gender and group. What I can infer from that figure is that the biggest difference in gaps between groups -i.e where parent's gap is larger than non-parent gaps- is on career satisfaction and community recognition. On research, although parents do present a larger gap, the non-parents' group also has a bias that is larger than on the other metrics. So this reinforces my doubt about the sample sizes of the two groups and how it affects the statistical results.

To be clear, I do not think this invalidates the results at all. The analysis has been thoroughly done. I would like to see some more robustness checks and details on this issue, and also on the interpretation of SUEST values.

22. P. 12 and table S11. The authors explain that work-family conflict can take the form of time, strain, and behavior-based conflict. They show the difference between groups for the work-family conflict, but they do not mention that for the three specific forms the non-parent group also showed significant differences between men and women. How can we interpret these apparently contradictory results?

23. P. 19 Regarding the policy recommendations, I would add that the metric-based evaluation, and especially the productivity-based career evaluations would harm mothers, as the paper has shown. So, qualitative evaluations that take into consideration parenthood and the individual conditions would also be very important to reduce gender biases.

---

## [Author Response]

Essential revisions:1. Why do "All" parent categories have higher ARPs and ARCs (for both men and women) compared to nonparents (Table S10)? Do female parents produce more publications and receive more citations than female non-parents? Do male parents produce more publications and receive more citations than male non-parents? This seems like an important point to discuss and one which is at odds with the author's overall thesis. Also, as mentioned below, Morgan et al. 2021 report that faculty of both genders who have children are slightly more productive than those who do not. This point should be discussed and put into context with the author's data/conclusions.

Thank you for the close attention to the results. We agree that the mean values of ARP and ARC look higher for parents than for non-parents. However, we did not compare the mean values between parents and non-parents directly due to the considerations of (1) the skewness in distributions and (2) the absence of control variables. Mean values are sensitive to extreme values and depend on the distributions of data. We plotted the distributions of ARPs and ARCs by gender, parenthood status, and career stage (see Author response image 1). We found that the values of ARPs and ARCs overall fall into normal distributions with close means and “long tails” across different genders and parent groups, but the tails are longer for the parent groups overall. Additionally, the effect of control variables is not considered in the direct comparison of means, which could be problematic.

Therefore, we used regression analysis to compare between parents and non-parents, controlling for a set of covariates (similar to other analysis in the project). As shown in *SI Appendix* Table S11, there is no significant difference between mothers and non-mothers in ARP, ARC, or ARCo. However, fathers do have higher ARP and ARCo values than non-fathers, but not ARC. The finding of higher ARP for fathers than non-fathers is consistent with that of Morgan et al. (2021). Yet, we did not find such a pattern between mothers and non-mothers, which is different from Morgan et al. 2021. Additionally, given the potential positive relationship between collaboration and productivity (Larivière et al., 2015), the relationship between the ARP values of fathers and non-fathers could also be related to their collaboration, in addition to fatherhood. We added the following text into the manuscript:

“Additionally, the mean values of the of ARP and ARC are generally higher for the parent group than the non-parent group for both genders (see Table Supplement Table S10). Yet, it should be noted that the direct comparison of means ignores the potential skewness in data distribution and the effect of control variables. Therefore, we further performed regression analysis to compare the parent and non-parent group (see Table Supplement Table S11). We found no significant difference between mothers and non-mothers in ARP, ARC, or ARCo. However, fathers do have higher ARP and ARCo values than non-fathers, but not ARP. The finding of higher ARP for fathers than non-fathers is consistent with that in Morgan et al. (2021). Yet, we did not find such a pattern between mothers and non-mothers, which is different from Morgan et al. (2021). Given the potential positive relationship between collaboration and productivity (Larivière et al., 2015), the relationship between the ARP values of fathers and non-fathers could also be related to their collaboration, in addition to fatherhood.”

**Author response image 1. sa2fig1:** Kernel density distributions of annual relative publications (ARP) and average relative citations (ARC) across parenting groups and genders, by career stages.

2. I did not extensively math check, but in areas where I did, the numbers were confusing. Unless I am misunderstanding where the numbers in the text are coming from.For example, on page 10,– "women's ARP is 0.28 units lower than men": On the graph the numbers are 2.17-1.79 = 0.38?– "women's ARC is 0.21 units lower than men": On the graph the numbers are 2.27-2.08 = 0.19?Or on page 11,– "ARP, ARC and ARCo are 0.35, 0.2 and 0.10 units…" On the graph the numbers are 2.27-1.82 = 0.45, 2.28-2.13 = 0.15 and 1.09-0.98 = 0.11?

We appreciate the comment. The number in statements like "women's ARP is 0.28 units lower than men" refers to the linear regression coefficient for the independent variable gender. We did not try to compare the mean values in these statements, knowing that would likely be skewed due to situations like extreme values, distributions, and control variables. To clarify the confusion, we have moved the 95% confidence intervals and the *p*-values closer to the coefficients. The revised text reads as follows:

“Our results further confirmed gender disparities in objective career achievements (See Figure 1 and Table Supplement Table S10). Regardless of parenthood status, women are outperformed by men in all three measures of objective career achievement. Specifically, women’s ARP is 0.28 (95% CI [-0.39, -0.17], p=0.000) units lower than men, women’s ARC is 0.21 (95% CI [-0.38, -0.05], p=0.013) units lower than men, and women’s ARCo is 0.08 (95% CI [-0.14, -0.01], p=0.023) unit lower than men. This echoes previous findings that women produce fewer publications (Astegiano et al., 2019; Besselaar and Sandström, 2017), receive lower citations (Larivière et al., 2013; Maliniak et al., 2013), and have fewer coauthors (Ductor et al., 2021) than their male counterparts.”

3. The author's discuss the "motherhood penalty". There is also a well-documented "fatherhood premium." How does this fit in with the author's results? Do the author's see evidence of a "fatherhood premium" in their data?

We are grateful for these questions. To answer these questions, we further compared the parent and non-parent group in terms of their objective and subjective career achievements (see *SI Appendix* Table S11). We did not find “fatherhood premium” in any of the three measures of subjective career achievements. However, we did find that fathers are more productive than non-fathers, showing some levels of fatherhood premium in research productivity. In the meantime, we also observed that fathers are more collaborative than non-fathers, which could be related to the higher productivity of fathers besides fatherhood, given possible causal effects between collaboration and productivity (Larivière et al., 2015).

4. Morgan et al. (https://www.science.org/doi/10.1126/sciadv.abd1996) surveyed 3064 tenure-track faculty in 450 departments in the US and Canada. What advances are made in the current datasets/analyses? How do the results compare? In addition to the point mentioned above about faculty of both genders who have children being reported as more productive than those without, Morgan et al. also reports that results are inconclusive as to whether long-term publication rates are affected by parenthood. Can the authors place their study and results into context here?

We appreciate that the reviewers brought up the important contribution by Morgan et al. (2021). Compared with their study, our study is different in many ways, including (but not limited to):

(1) Morgan et al.’s study focused on the gender gap in productivity due to parenting, while our study examines gender gaps in a broader spectrum by covering both objective (productivity, citation, and collaboration) and subjective (research satisfaction, career satisfaction and community recognition) career achievements.

(2) Morgan et al.’s study focused on three disciplines (computer science, business, and history), while our study examines all disciplines (as covered in Web of Science).

(3) Morgan et al.’s study analyzed productivity, position satisfaction and their relationships with parental leave policy, while our study focused on the role of family (especially partners) in shaping the gender disparities we observe.

The difference between the Morgan et al.’s study and this project goes beyond that. But we think our research aligns with Morgan et al.’s study in some way and complements it with additional findings by focusing on a wider spectrum of disciplines, assessing both subjective and objective career achievements, and concentrating on the role of partners and families.

Regarding productivity, the Morgan et al.’s study found that mothers’ productivity drops at a higher rate than fathers’ right after the birth of their children. They also found that parents are overall slightly more productive than non-parents regardless of gender. We did also find fathers are more productivity than non-fathers overall, but we did not find mothers are more productive than non-mothers. As mentioned in our responses to comments #1 and #3, the higher number of collaborators fathers have than non-fathers might also be related to fathers’ higher productivity. Future research might consider looking into the causality between these two factors.

As for the long-term impact of parenthood on productivity, because we did not collect the time of each childbirth for parents, our analysis on the long-term impact of parenthood on productivity was limited. As an approximation, we analyzed the productivity gender gap in both parent and non-parent group based on career stage (see Table Supplement Table S10), based on the assumption the career stage and length is reflective of the time of childbirth. We are aware that this might not be the case for many academics. Overall, our result shows that mothers are less productive than fathers across early, middle, and late career. But women and men in the non-parent group do not show this pattern. Therefore, the relationship between parenthood and gender gaps in productivity seems to go beyond early or mid-career and exists across all career stages.

5. Were there any differences for men/women parents/non-parents observed between different fields within academia? Are the conclusions the same regardless of discipline? Are the disparities the same regardless of discipline?

We thank the reviewers for bringing up these important questions. We compared the subjective and objective career achievements between genders by disciplinary area. Regarding objective career achievements, we found that fathers are more productive than mothers in medical sciences, and natural sciences and engineering, but not in arts and humanities, and social sciences. Because we rely on data from Web of Science, it is likely that academics productivity is not fully captured for arts and humanities, and social sciences due to the known coverage gap in these two areas by Web of Science. Regarding subjective career achievements, women are more likely to be less satisfied with their research than men in arts and humanities, medical sciences, and natural sciences and engineering. Mothers are more likely to have lower research or career satisfaction in all four disciplinary areas, and less likely to feel recognized by their scholarly communities in natural science and engineering, and medical sciences.

While the conclusions for some disciplinary areas are slightly different from others, this does not necessarily indicate that our previous conclusions are invalid. These differences are expected due to variances in scholarship practices among those disciplinary areas.

**Author response image 2. sa2fig2:** Arts and humanities.

**Author response image 3. sa2fig3:** Medical sciences.

**Author response image 4. sa2fig4:** Natural sciences and engineering.

**Author response image 5. sa2fig5:** Social sciences.

6. The authors do not distinguish between respondents who currently have childcare responsibilities (i.e., children < 18 currently residing with them) versus those who have ever had children (i.e., children > 18, non-custodial parents, etc.). The authors also do not distinguish between same-sex and opposite sex parents, where gender roles and expectations may differ. These should be raised in the Limitations section of the manuscript, as the current analyses focus on a model of a traditional, opposite-sex, nuclear family.

We completely agree with the issues brought up by the reviewers. We have added the following text to our *Limitation* section per reviewers’ suggestions.

“The current study is also limited by the fact that the survey did not collect data to distinguishing same-sex and opposite-sex partners, where gender roles and expectations may differ. Additionally, the current study lacks data on the types of childcare responsibilities by parents, such as whether they are custodial parents or non-custodial parents, and whether children live with them. To further understand how childcare responsibilities contributes to gender gaps in academia, data and analyses based on the factors mentioned above will be critical for understanding the true mechanisms and possible policy implications for address gender inequalities in the scientific workforce.”

7. Data needs to be fully referenced within the text and shown if referenced. For example, on page 7 of the Results section, the authors refer to women in academia being more likely to have fewer children (presumably as compared with men?). There is not a reference for a data table here and the data does not appear to be in the supplemental tables? The authors then state "This trend is different from the US general population, where more adult women have children and have more children than men." (REF). Do the authors mean primary caretaking responsibilities of children or children that reside with them?

We thank the reviewers for bringing this up. In the statement mentioned by the reviewer, we did not reference a table because it only involved one single regression analysis. We have appended the details of this regression to Table S6 and added the reference to the text.

As for the statement "This trend is different from the US general population, where more adult women have children and have more children than men", it was based on the 2017 Survey of Income and Program Participation by the U.S. Census Bureau. Based on the survey description, we don’t believe they differ in terms of childcare responsibilities. Our survey did not distinguish parents in this manner either, which we acknowledge as a limitation of our study. Thus, this statement simply commented on the number of children they have. We have revised the sentence as follows.

This trend is different from the general population in the United States, where the percentage of adult women who have children is higher than that of adult men, and the average number of children mothers have is higher than that of fathers (41; 43).

8. On page 11, last paragraph, the authors discuss results around career stage. This needs a reference to the data table.

We have added Table S10 to this paragraph as the reference table.

9. On page 7, the authors state: "Among those without children, women (59.9%) are more likely than men (43.2%) to report that they chose to be childfree due to career considerations…". If I understand correctly, the survey question was: "Is the number of children (include 0) you currently have related to your career considerations (more or less)?" The interpretation is cause and effect (women chose to be childfree due to career) whereas the survey question asks if these were related considerations (which could include additional factors?).

We highly appreciate the comment, which we completely agree with. We have changed the mentioned texts to the following to avoid causal implications:

“Among those without children, women (59.9%) are more likely than men (43.2%) to report that their choices to be childfree were related to career considerations (OR=2.10, 95%CI [1.69, 2.62], p=0.000; see Table Supplement Table S7), while controlling for discipline, career stage, and race.”

10. In the Introduction, the authors use the term "non-binaries" to refer to individuals whose self-reported gender identity on the survey was "Other". Please consult the APA guidelines for bias free language: https://apastyle.apa.org/style-grammar-guidelines/bias-free-language/gender

We highly appreciate the reviewers for the comment and suggestions. We’d like to clarify that we classified those who chose the “Other” category and added text comments that imply them to be in the non-binary category as non-binaries. Such comments may include (but are not limited to) keywords such as genderqueer, agender, bigender, gender-fluid, non-binary, gender-nonconforming, gender-neutral, and gender-creative.

11. In the Introduction, the authors state, "The multigenerational gender role beliefs, which characterize men and women with a simplistic set of features, may direct women academics toward women's normative roles in the family (REFs). Some women academics may become 'maternal gatekeepers" who are reluctant to relinquish family responsibility" (REFs). This seems to stray from the facts and into speculative territory not supported by the author's data.

Thank you for your comments on the Introduction part. We agree that these statements are a bit stray from the facts and not supported by our data. We thus removed the statements from our manuscript.

12. In the Introduction, the authors state, "Although partner support is believed to be more critical to women's career success than to men, especially for those in parenthood, women's demand for partner support is less satisfied" (REFs). What does this mean? Women are more dependent on partner support for career success regardless of parenthood? Is this supported by data? Men need less partner support even in parenthood? Doesn't the author's data show male academic parents receive more time support, etc.?

We thank the reviewers for the comment. We have modified our text to the following to avoid confusions:

“Although partner support has been found to help reduce mothers’ work-family conflict to an extent greater than fathers’, women’s demand for partner support is often less satisfied (Adams and Golsch, 2021; Dickson, 2020).”

13. This sentence between page 8-9 is confusing: "While the satisfaction over research is significantly different between mothers and fathers, our results show no significant gender differences in the career satisfaction of parents…therefore, it is likely that women consider more on other gains (e.g., emotional gains) from teaching and service while assessing their overall career satisfaction, while men emphasize research." What do the authors mean by "women consider more on other gains (e.g., emotional gains)" and is this data-driven or speculation? And what is meant by "men emphasize research"? It seems a stretch to attribute motivation / feelings to the author's data.

We thank the reviewers for these comments. We agree that the original statement is a little stretchy from our data. We have changed the related text to the following to avoid possible ambiguities and conjecture.

While the satisfaction over research is significantly different between mothers and fathers, our results show no significant gender difference in the career satisfaction of parents (OR=0.89, 95%CI [0.77,1.03], p=0.129). Furthermore, of those women who expressed satisfaction over their career achievements, 13.6% were not satisfied with their research achievements. The number is 8.9% for men (see Table Supplement Table S9). Research, teaching, and service being the three primary duties for many tenure-track faculty, it has been found that teaching and service are more heavily loaded on women than on men (Bellas, 1999; Guarino and Borden, 2017). If teaching and service are indeed loaded heavier on women, then mothers’ lower satisfaction rates on research could be related to the extra amount of time and effort they put on these aspects, on top of mothers’ already

14. What does this sentence on page 13 mean, "women are usually disproportionately associated with technical work in research" (REF)?

We thank the reviewers for the question. We meant to cite the Macaluso et al. (2016) study’s finding that women were significantly more likely to be associated with tasks such as performing experiments, especially in laboratories. Given that experiments might not necessary be the technical support in our study, we removed the statement to avoid any possible stretch of our results.

15. In the Discussion section, the authors state, "the perceived negative impact of parenting is one of the major systematic barriers for women's self-selecting attrition from academia…" Can the authors support the claims of self-attrition and self-attrition due to a perceived negative impact of parenting? This seems to imply that women are deciding it's too hard to have children and an academic career, and then deciding to leave academia. Is there data to support this? This also entirely ignores bias/discrimination against women and mothers, which is well documented in academia.

We thank the reviewers for the questions and comments. By the statement, "the perceived negative impact of parenting is one of the major systematic barriers for women's self-selecting attrition from academia, contributing to the "leaky pipeline" in academia (Anders, 2004)”, we meant to express our agreement with the findings in the Anders’s (2004) study, where they found systemic barriers associated with parenting discourage women from pursuing academic careers. We agree with the reviewers that parenting related issue is only one reason for the “leaky pipeline”, and many issues such as bias and discriminations against women and mothers also contribute significantly to gender inequalities in science. This was also why we emphasized that “the perceived negative impact of parenting is one of major systemic barriers…”. To avoid possible confusions, we rephrased the statement as follows:

“The perceived negative impact of parenting could intensify gender inequalities, as previous research (Anders, 2004) found it was one of the major systemic barriers for women's self-selecting away from academia, contributing to the academic "leaky pipeline".”

Reference:

Anders, S. M. van. (2004). Why the Academic Pipeline Leaks: Fewer Men than Women Perceive Barriers to Becoming Professors. Sex Roles, 51(9), 511–521. https://doi.org/10.1007/s11199-004-5461-9

16. In the Discussion section, while I don't believe the authors intended this statement to be offensive, some portions may come across that way: "Women in academia…tend to slough off the burden of children… Given the motherhood penalty, it is unsurprising that many women stay in academia by paying the price of being childfree…" In particular, this places value judgements around having children or not having children. It also seems unsupported by the data.

We highly appreciate the comment. We have corrected our language in the text to be:

“On the other hand, we found that women are less likely than men to have children and have fewer children. The insignificant differences between non-fathers and non-mothers in their productivity, citation, and collaboration imply that parenthood contributes to the gendered differences in these three aspects for parents. Given the motherhood penalty, it is unsurprising that academic women have fewer children and are less likely to have children than men, which our results suggested as being related to career considerations: Our results show non-mothers are more likely than non-fathers to report their decision to be childfree was related to career considerations and mother academics are more likely than father academics to report that the lower number of children they have was also related to career considerations.”

17. In the Discussion section, the authors discuss the finding that "mothers are more likely to receive financial and technical support from their partners." By stating "It reflects the entrenched gender division of household labor that expands with the new childcare task: Mothers are recognized, and even morally self-recognized, as the babysitters and fathers are the breadwinners" (REF). "As a social norm…fathers bear the financial responsibility. In contrast, fathers compensate mothers' loss due to extra childcare responsibility by offering financial and technical support." Again, can the authors draw this conclusion from their data?

We highly appreciate the comments. In the Discussion section, by stating "It reflects the entrenched gender division of household labor that expands with the new childcare task: Mothers are recognized, and even morally self-recognized, as the babysitters and fathers are the breadwinners (Hauser, 2012). As a social norm, it is more common that mothers sacrifice their working time, emotionally support fathers' work, and understand their decisions because fathers bear the financial responsibility.", we are trying to connect our findings to potential reasons behind that. We have changed our statements to the following:

“On the contrary, mothers are more likely to receive financial and technical support from their partners. Mothers’ higher likelihood of receiving financial support might be related to the entrenched gender division of household labor, where fathers are more likely to be the breadwinners and bear financial responsibilities (Hauser, 2012).”

18. In the Discussion section, the authors again make reference to "high attrition rates of mothers in the scientific workforce due to lower satisfaction rates." Can the authors backup the idea of higher attrition rates of mothers compared with non-mothers in the scientific workforce, and also that the reasons for attrition are due to lower satisfaction rates (i.e., self-selection)?

We highly appreciate the comments. Our data do not support the claims directly. We intended to cite previous research and connect with our findings. We have changed the corresponding text as follows:

“Moreover, the higher levels of work-family conflict experienced by mothers also worsen the gender gap in satisfaction over research and perceived recognition by parents, which are highly related to women's turnover intention (Watanabe and Falci, 2016). Reducing mothers' work-family conflict and increasing their general support could thus reduce the gender gap in academics' perceived career achievement. These mediating variables indicate the directions for mitigating the negative impacts of motherhood: Partner support in the forms of time, emotional, and decision support, and other collective efforts to reduce the time-based, strain-based, and behavior-based conflict for mothers.”

19. P.6. I am happy to see that the authors included non-binary people in their survey. I also understand that the small sample makes it impossible to consider NB into the regression models. Nevertheless, it would be very important to include their responses somehow. A qualitative analysis of the 28 responses from NB people after the quantitative analysis of men and women, to see where they locate within the more general results would be a very interesting complementary analysis.

We agree with the reviewers that the inclusion of non-binary respondents is critical, but any regression analysis will not be feasible due to the limited sample size. Here is the descriptive analysis regarding the 28 no-binary respondents.

In total, 28 non-binary participants responded to our survey, of which 18 ever committed partnership. Seven of them have children, among which six (85.7%) indicated that having children has a negative impact on their career, and all seven (100%) respondents reported that the number of children they have is related to career considerations.

Regarding the three measures of subjective career achievement, 5 out of 7 (71.4%) parents are not satisfied with their research achievements, as compared to 4 out of 11 (36.4%) non-parents. For career satisfaction, 4 out of 7 (57.1%) parents expressed disagreement, as compared to 2 out of 11 (18.2%) non-parents. For scholarly recognition, 1 out of 6 (16.7%) parents expressed disagreement, as compared to 2 out of 11 (18.2%) for non-parents.

Regarding the three measures of objective career achievement, the ARP is 1.34 for parents and 1.36 for non-parents. The ARC is 1.58 for parents and 1.79 for the non-parents. The ARCo is 0.92 for the parents and 0.98 for the non-parents.

The limited descriptive analysis shows that parenthood is also related to non-binary respondents’ career achievements. However, given the small sample size, the above comparisons may lack statistical power and therefor be interpreted with limited generalizability.

20. It is not clear to me why the authors decided to send the survey to 25% of the population. If there was a technical limitation, it would be good to state it. Besides this, there is a 9% response rate. It would be helpful to see some robustness checks o whether that 9% is representative of the population or not. Some comparative statistics between respondents and non-respondents would help to disclose some potential self-selection biases.

We thank the reviewers for the suggestions. We took a random sample consisting of 25% (99,168) of the population mainly because the version of Qualtrics (the tool we used to distribute the survey) had a weekly limit (10,000) for the number of email addresses that we could send the survey to. We thus decided to take a sample so that we could distribute the survey in a reasonably short period of time. Our response rate is lower than the Morgan et al.’s study, but a bit higher than some other studies (Ni et al., 2021).

To evaluate the representatives of our analytical sample, we compared the gender and discipline area compositions in the population, surveyed group, respondents, and final analytical sample. The distributions are shown in the updated Table S1. Please note that the gender categorization in the following table was decided using the method by Larivière et al. (2013), which is different from the gender classification in our analyses (based on self-reported data from the survey).

References:

Larivière, V., Ni, C., Gingras, Y., Cronin, B., and Sugimoto, C. R. (2013). Bibliometrics: Global gender disparities in science. Nature, 504(7479), 211–213. https://doi.org/10.1038/504211a

Ni, C., Smith, E., Yuan, H., Larivière, V., and Sugimoto, C. R. (2021). The gendered nature of authorship. Science Advances, 7(36), eabe4639. https://doi.org/10.1126/sciadv.abe4639

21. The sample size of parents and non-parents is different, with a larger number of parents to non-parents (roughly 80% to 20%). Statistical measures are usually dependent on sample size, with larger sample sizes having more rejection power. Is it possible that the differences seen between groups, no significant bias on non-parents and significant bias on parents is just and statistical artifact of the sample sizes? The authors could try to check if this is the case using bootstrapped samples of equal size, or also consider directly the absolute values of the metrics to see if there is an important difference between the two groups -besides the statistical difference.Further on the previous point. On Figure 1A the authors show the different subjective satisfaction measures by gender and group. What I can infer from that figure is that the biggest difference in gaps between groups -i.e where parent's gap is larger than non-parent gaps- is on career satisfaction and community recognition. On research, although parents do present a larger gap, the non-parents' group also has a bias that is larger than on the other metrics. So this reinforces my doubt about the sample sizes of the two groups and how it affects the statistical results.To be clear, I do not think this invalidates the results at all. The analysis has been thoroughly done. I would like to see some more robustness checks and details on this issue, and also on the interpretation of SUEST values.

We appreciate the comments. We agree that the relatively small sample size of the non-parent group and the difference of sample sizes could potentially raise concerns. Therefore, as a robustness test of our results, we used non-parametric bootstrapping to estimate the 95% bias-corrected confidence intervals of our models across the non-parent and parent groups. The test is performed 5,000 times by resampling with replacement from each group respectively to obtain a series of new samples of the same sample sizes. If the 95% bias-corrected confidence interval does not contain zero, we will consider the result as significant. Our bootstrap results (see Author response table 1) show that overall, the results are consistent with the results in our original manuscript.

**Author response table 1. sa2table1:** The bootstrap results for subjective and objective career achievements – coefficients and 95% bias-corrected confidence interval (CI).

Subjective career achievement	Coef.	95% bias-corrected CI	Objective career achievement	Coef.	95% bias-corrected CI		
		Lower	Upper			Lower	Upper
**Non-parent**							
Research satisfaction	-0.062	-0.317	0.185	ARP	0.016	-0.148	0.174
Career satisfaction	0.052	-0.227	0.334	ARC	-0.321	-0.765	0.032
Community recognition	-0.05	-0.333	0.226	ARCo	0.025	-0.109	0.141
**Parent**							
Research satisfaction	-0.332	-0.457	-0.208	ARP	-0.349	-0.486	-0.216
Career satisfaction	-0.116	-0.25	0.035	ARC	-0.196	-0.386	-0.01
Community recognition	-0.308	-0.454	-0.162	ARCo	-0.102	-0.174	-0.031
**Coefficient difference (non-parent – parent)**							
Research satisfaction	0.271	-0.016	0.549	ARP	0.365	0.152	0.572
Career satisfaction	0.168	-0.149	0.476	ARC	-0.126	-0.599	0.273
Community recognition	0.259	-0.063	0.558	ARCo	0.126	-0.02	0.263

The p values of SUEST are used to show whether the women/men odds ratio value in the parent group is significantly different from that in the non-parent group. If the *p*-value is below the predetermined significance level, we will accept the null hypothesis that the effect of gender (women/men) on the dependent variable is significantly different in the regression models for the parent and non-parent groups. We also performed bootstrapping to check the robustness of the results, in which we obtained 5,000 cross-group coefficient differences for gender (women/men) between the non-parent and parent groups to draw empirical *p* values. We found that the bootstrapping results are consistent with the results using the SUEST methods.

22. P. 12 and table S11. The authors explain that work-family conflict can take the form of time, strain, and behavior-based conflict. They show the difference between groups for the work-family conflict, but they do not mention that for the three specific forms the non-parent group also showed significant differences between men and women. How can we interpret these apparently contradictory results?

We thank the reviewers for their comments. We agree that there are noticeable differences between women and men in the non-parent group concerning time, strain, and behavior-based conflict. However, the levels of conflicts at all three forms are different between the non-parent and parent group (see Table S12):

The percentage of non-parents experiencing work-family conflict: time-based conflict (women=39.1% vs. men=27.1%), strain-based conflict (women=16.1% vs. men=14.1%), and behavior-based conflict (women=8.8% vs Men=5.1%).The percentage of parents experiencing work-family conflict: time-based conflict (women=54.7 % vs. men=39.0 %), strain-based conflict (women=30.8% vs. men=20.6%), and behavior-based conflict (women=11.5% vs men=7.9%).

Overall, the level of conflicts is much lower in the non-parent group than in the parent group, although it still shows difference between non-parent women and men. This may further imply that women are more likely than men to experience the three types of conflict, but having children may exacerbate the gap. Therefore, we do not think these results are contradictory. Instead, this further implies that children matter a lot in the difference in work-family conflict experienced by women and men.

23. P. 19 Regarding the policy recommendations, I would add that the metric-based evaluation, and especially the productivity-based career evaluations would harm mothers, as the paper has shown. So, qualitative evaluations that take into consideration parenthood and the individual conditions would also be very important to reduce gender biases.

We highly appreciate the suggestions. We have added the following text to the Discussion section on Page 19.

“Additionally, given the observed gender gaps in research productivity, the current metric-based evaluation, especially those based on productivity, will undoubtedly hurt mothers more. Therefore, additional metrics and qualitative evaluations that take into account parenthood and individual conditions would be critical to reduce gender biases in academic settings.”